# Hierarchical Multi-Omic CLIP for Missing-Modality Imputation & Transfer Learning in Blood Cancers

**Leonardo P. A. Biral**
Department of Computational Biology & Bioinformatics, Duke University
Durham, NC, USA
leonardo.biral@duke.edu

**Dennis Owusu**
Department of Computational Biology & Bioinformatics, Duke University
Durham, NC, USA
dennis.owusu@duke.edu

**Sandeep Dave**
Department of Medicine, Duke University
Durham, NC, USA
sandeep.dave@duke.edu

## Abstract

Blood cancers are a major public health burden, affecting more than 10 million people worldwide. Genomic profiling has improved patient outcomes, but machine learning models still struggle to generalize because multi-omic cohorts are sparse, often lack entire modalities, and exhibit strong out-of-distribution (OOD) shifts across institutions and rare diagnoses. Here we introduce BLOOM-HiCLIP, the first hierarchical multi-omic CLIP framework for blood cancer, trained on the largest multi-omic blood cancer cohort to date of over 8,200 tumors spanning 165 diagnoses. BLOOM-HiCLIP leverages biological foundation models to learn taxonomy-consistent patient representations by mapping RNA and DNA-derived omics modalities to a shared latent space with high cosine similarity between imputed and true embeddings. The resulting joint embedding space supports strong cross-modal retrieval (94.8% Recall@5) and effectively captures blood cancer's hierarchical structure, outperforming a matched InfoNCE-trained ablation model. Notably, BLOOM-HiCLIP freezes all foundation encoders, training only lightweight projection heads and pooling parameters (39.5M total), achieving strong performance with 33.4x fewer parameters than costly end-to-end finetuning. When transferred to a hierarchical diagnostic model, these contrastive embeddings achieve strong discrimination across the blood cancer hierarchy (98.1% cell-of-origin micro-averaged AUROC) outperform baselines in fine-grained diagnosis across 29 subtypes on both in-distribution (34.8% top-1 accuracy) and OOD settings. These embeddings also effectively transfer to risk stratification, achieving an overall survival c-index of 0.788. BLOOM-HiCLIP demonstrates that hierarchical contrastive alignment can turn heterogeneous, incomplete multi-omic data into imputation-ready representations that reliably transfer to clinically relevant prediction tasks in blood cancer.

## 1 Introduction

Blood cancers impose a substantial global health burden, representing roughly 10% of all cancers and affecting more than 10 million individuals worldwide (National Cancer Institute & Program, 2025). Their clinical management is complicated by their high diversity, with over 160 distinct en-

tities organized in a multi-level taxonomy within the 5th Edition of the WHO Classification System for Hematolymphoid Tumors (WHO5) (Li, 2022). The integration of multi-omic data has improved the characterization of these malignancies, but clinically collected multi-omic cohorts are often incomplete as omics profiling remains expensive and is not uniformly applied (Flores et al., 2023; Hornung et al., 2023). As a result, many patients lack entire modalities and different institutions exhibit systematic differences in sequencing quality and availability (Leek et al., 2010).

CLIP-style contrastive learning offers a compelling mechanism to address missingness and heterogeneity by aligning paired views of the same underlying entity in a shared embedding space (Radford et al., 2021). In biology, CLIP-based methods have begun to demonstrate the utility of contrastive objectives, particularly in single-cell settings where paired measurements across modalities are increasingly available (Xiong et al., 2023; Ebrahimi et al., 2025; Gossi et al., 2023). To our knowledge, no CLIP-style models have been developed specifically for hematologic malignancies, likely reflecting both the lack of large multi-omic blood cancer cohorts and the unique modeling challenges posed by these diseases. Namely, CLIP needs to be robust to variable-length molecular event sets (e.g., few fusions versus dozens of variants), handle institution-dependent modality missingness, and learn representations that respect clinically-defined hierarchies such as WHO5.

In parallel, biological foundation models have made it feasible to extract rich embeddings from individual modalities without training large encoders from scratch. This includes transformer models for bulk transcriptomics (Gélard et al., 2025) and nucleotide sequences (Dalla-Torre et al., 2024), protein (Lin et al., 2023) and biomedical text (Gu et al., 2021) language models, and gene-set foundation models that represent pathway-level biology (Clarke et al., 2025). However, leveraging these pretrained components for clinical multi-omics is nontrivial. A practical system must integrate heterogeneous foundation embeddings into a unified patient representation, handle sparse, variable-length event sets, remain robust under modality missingness, and produce representations that can be transferred to clinically relevant tasks all while respecting disease taxonomy and maintaining reasonable training costs.

We address these gaps with Blood cancer Learning via Omics and Ontology-aware Modeling with Hierarchical CLIP (BLOOM-HiCLIP), the first hierarchical multi-omic CLIP framework designed for heterogeneous blood cancer cohorts with incomplete profiling. BLOOM-HiCLIP is trained on the Atlas of Blood Cancer Genomes (ABCG) dataset, the largest multi-omic cohort of hematological malignancies to date, comprising 8,281 genomically-profiled pre-treatment tumors spanning 28 institutions and 165 disease subtypes (Love et al., 2021). It aligns RNA and DNA-derived patient embeddings from foundation models into a shared latent space, enabling principled missing-modality imputation. This model handles variable-length event sets by representing molecular event types as token sequences, using CLS-pooling to summarize them as fixed-dimensional patient embeddings. By freezing foundation encoders and training only lightweight projection heads, BLOOM-HiCLIP achieves strong performance with a fraction of the parameters required for end-to-end fine-tuning. BLOOM-HiCLIP further incorporates a hierarchy-aware contrastive objective that encourages the embedding geometry to be consistent with the WHO5-defined ontology, using soft targets derived from taxonomic distance rather than treating all non-matching patients as equally negative. The resulting embeddings provide a stable representation for downstream transfer to two clinically-impactful tasks: hierarchical diagnosis and survival prediction.

**Contributions.** Our contributions are as follows:

1. BLOOM-HiCLIP, the first hierarchical multi-omic CLIP framework for blood cancer, incorporating a taxonomy-aware soft-target contrastive objective that respects the WHO5 disease hierarchy

2. A tokenization strategy that represents variable-length molecular event sets as sequences, enabling patient-level embeddings via CLS-pooling

3. A parameter-efficient architecture that integrates five biological foundation models while training only 39.5M parameters (33.4x fewer than full fine-tuning)

4. Demonstration that the learned embeddings support strong cross-modal retrieval and effective missing-modality imputation

5. Transfer learning results showing the learned embeddings improve hierarchical diagnosis and survival prediction

## 2 METHODS

### 2.1 PIPELINE OVERVIEW

We aligned the ABCG cohort (n=8,281; 165 diagnoses) to the WHO5 taxonomy and processed each case through standardized pipelines to generate both RNA (bulk expression and fusions) and DNA (genomic variants) derived modalities (Figures 1a-b). We trained BLOOM-HiCLIP on the subset of patients with paired RNA and DNA modalities, learning a shared cross-modal embedding space whose patient-level outputs we refer to as Latent Alignment Taxonomy-guided Trans-omic Integrative Contrastive Embeddings (LATTICEs). Post-training, we use available omics inputs to represent all patients in LATTICE space for transfer learning. Separately, tabular clinical variables such as demographics, stage, family history, and complete blood counts were encoded with task-specific clinical encoders for diagnosis and prognosis. The resulting clinical embeddings were concatenated with LATTICE and passed into the downstream target models.

### 2.2 MODALITY AVAILABILITY & EVALUATION COHORTS

As typical in clinical cohorts, modality availability was heterogeneous based on sequencing quality with some patients missing either RNA or DNA-derived features while most (65.9%) had both (Figure 1c). We designed our evaluation splits to explicitly test model generalizability to samples from OOD institutions and diagnoses. We first designated three institutions as held-out OOD sites. Within ID institutions, we defined an ID diagnosis set from samples with diagnoses containing at least 50 paired RNA-DNA cases. This procedure led to four evaluation cohorts:

1. **IIDI** (institutions ID, diagnoses ID): n=5,296; 29 diagnoses; 25 institutions
2. **IIDO** (institutions ID, diagnoses OOD): n=2,115; 132 diagnoses; 22 institutions
3. **IODI** (institutions OOD, diagnoses ID): n=555; 28 diagnoses; 3 institutions
4. **IODO** (institutions OOD, diagnoses OOD): n=315; 82 diagnoses; 3 institutions

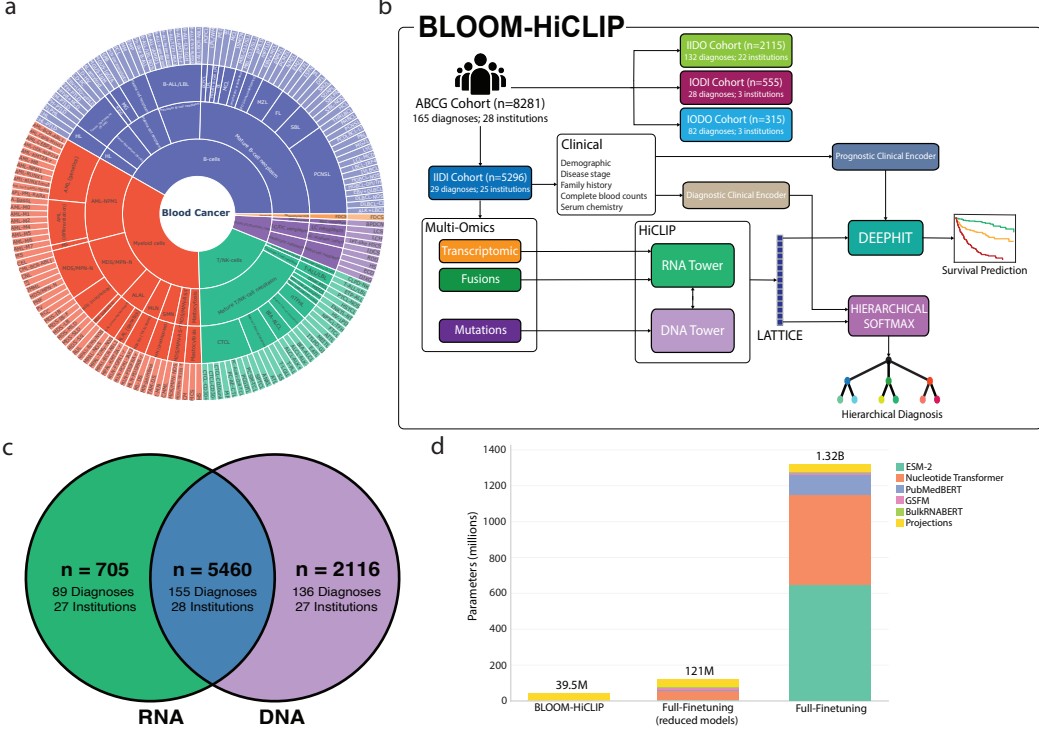

Figure 1: (a) WHO5 hierarchy. (b) BLOOM-HiCLIP schematic. (c) Modality missingness Venn diagram. (d) Parameter counts for BLOOM-HiCLIP vs. full finetuning.

## 2.3 FOUNDATION ENCODERS & PARAMETER-EFFICIENT TRAINING

The foundation encoders used in BLOOM-HiCLIP with their respective parameter counts are listed in Table S1. Rather than end-to-end finetuning these large encoders, we freeze them and train only the projection heads, pooling modules, and 3 scalar parameters. As a result, BLOOM-HiCLIP has 39.5M trainable parameters, 33.4x fewer than fully finetuning all foundation encoders and 3.1x fewer than finetuning the smallest version of each foundation encoder (Figure 1d).

In addition to foundation embeddings, each patient has structured tabular features summarizing metadata such as sequencing QC and number of events. Fusion and variant events also have event-specific metadata such as read support, fusion partner genes, and variant deleteriousness.

## 2.4 BLOOM-HiCLIP ARCHITECTURE & HIERARCHICAL CONTRASTIVE TRAINING

BLOOM-HiCLIP is a two-tower, patient-level CLIP model that maps RNA-derived and DNA-derived patient views into a shared $d$-dimensional latent space (LATTICE) (Figures 2a-b). Each tower converts a variable-length set of molecular events into a fixed-dimensional patient embedding via modality-specific tokenization followed by a transformer encoder. Specifically, we prepend a learnable `[CLS]` token to the variable-length token sequence, add learned token-type and positional embeddings, and pass the result through a transformer. The final hidden state of the `[CLS]` token serves as the patient embedding. Sequences are padded with `[PAD]` tokens, assigned the `PAD` token-type id, and masked during self-attention. The two towers have independent parameters, allowing each modality to learn specialized representations before alignment. We describe each tower's architecture and the overall training protocol in detail in the Appendix.

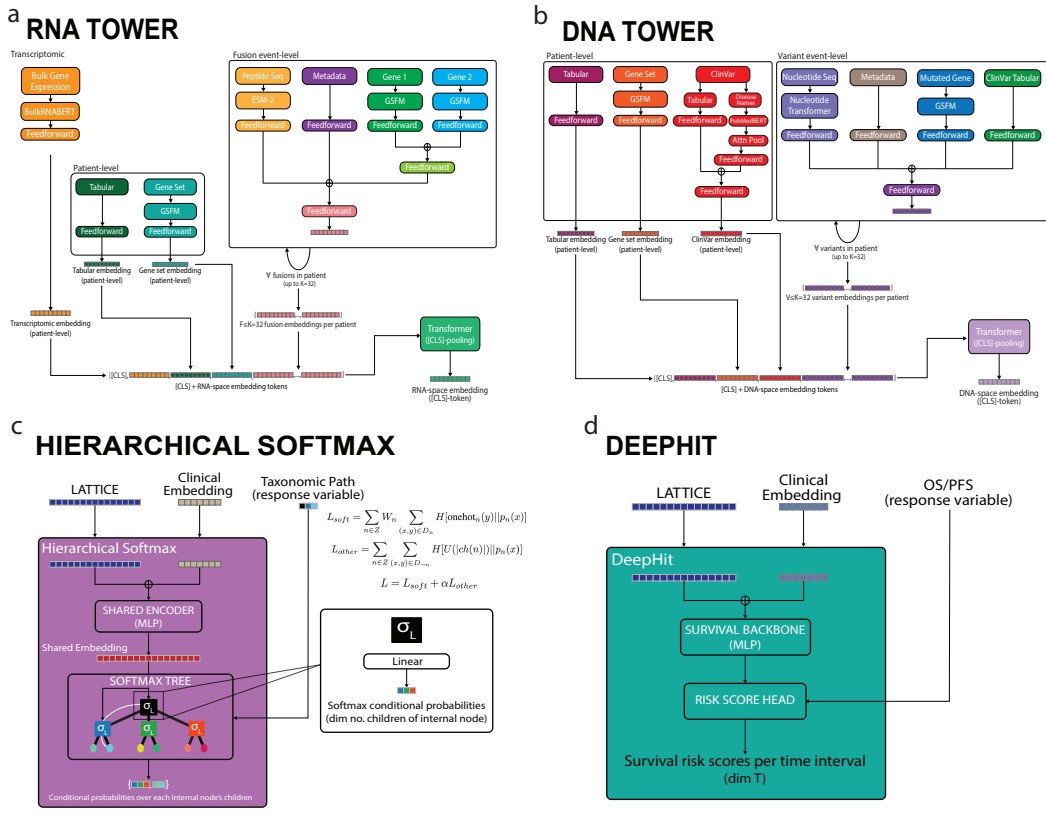

Figure 2: (a) RNA tower, (b) DNA tower, (c) HSM, and (d) DeepHit schematics.

## 2.4.1 HIERARCHICAL SOFT-TARGET CONTRASTIVE OBJECTIVE

Given a minibatch of paired patients $\{(x_i^{\mathrm{RNA}}, x_i^{\mathrm{DNA}})\}_{i=1}^{B}$, the model outputs embeddings $\{z_i^{\mathrm{RNA}}\}_{i=1}^{B}$ and $\{z_i^{\mathrm{DNA}}\}_{i=1}^{B}$, which are $\ell_2$-normalized. Similarities are computed by scaled dot

product with $s = \exp(\tau)$ as a learnable temperature parameter:

$$\ell_{ij} = s \cdot \langle z_i^{\text{RNA}}, z_j^{\text{DNA}} \rangle,$$

To incorporate diagnostic structure, we define a distance metric $d(i,j)$ using patient labels at four levels (cell-of-origin (COO), supercategory 2, supercategory 3, and diagnosis) based on the deepest common ancestor (DCA) distance between the diagnosis labels of $i$ and $j$ in the WHO5 taxonomy:

$$d(i,j) = \begin{cases} 1 & \text{same diagnosis} \\ 2 & \text{same supercategory 3, but different diagnosis} \\ 3 & \text{same supercategory 2, but different supercategory 3} \\ 4 & \text{same COO, but different supercategory 2} \\ \infty & \text{otherwise,} \end{cases}$$

We convert distances into nonnegative weights via exponential decay with learnable parameter $\alpha$:

$$W_{ij} = \mathbf{1}[d(i,j) < \infty] \cdot \exp\{-\alpha \cdot \max(d(i,j) - 1, 0)\},$$

Our true paired match weight $W_{ii}$ is a learned parameter initialized at 10 and parameterized so it always stays $> 1$. We row-normalize these weights into a soft-target distribution $P_{ij}^{\text{RNA}\rightarrow\text{DNA}} = W_{ij} / \sum_k W_{ik}$ (and analogously for the reverse direction using $W^\top$). The final loss is a symmetric soft-target InfoNCE averaged over batch elements:

$$\mathcal{L} = \tfrac{1}{2}\left(-\sum_j P_{ij}^{\text{RNA}\rightarrow\text{DNA}} \log \text{softmax}(\ell_{i\cdot})_j - \sum_i P_{ji}^{\text{DNA}\rightarrow\text{RNA}} \log \text{softmax}(\ell_{\cdot j})_i\right),$$

### 2.4.2 BLOOM-HiCLIP EVALUATION

To isolate the contribution of our hierarchical soft-target objective, we trained a matched InfoNCE ablation with identical data splits, architecture, and hyperparameters to BLOOM-HiCLIP, but replace the hierarchical soft-target loss with standard InfoNCE. All metrics are computed on each evaluation split for both retrieval directions (RNA→DNA and DNA→RNA) and are reported either per-direction or averaged across directions. Our evaluation metrics are described in the Appendix.

## 2.5 TARGET MODELS

LATTICEs are concatenated with task-specific clinical embeddings and fed into dedicated target models for hierarchical diagnosis and survival. We exclude clinical variables during BLOOM-HiCLIP training because their relevance is task-dependent (e.g., blood counts for diagnosis vs. age for prognosis). We expand on target model protocol in the Appendix. We compared baselines to LATTICE-augmented models across 5 random seeds, which quantifies the gain from LATTICE and each modality's contribution.

### 2.5.1 HIERARCHICAL SOFTMAX: HIERARCHICAL DIAGNOSIS CLASSIFICATION

A hierarchical softmax model (HSM) predicts along the WHO5 lineage by first mapping inputs through an MLP into a shared representation and then traversing a WHO5-aligned softmax tree where each internal node has its own linear classifier outputting a conditional distribution over its children (Figure 2c). Post-training, we calibrate node-specific confidence thresholds and perform top-down inference with early stopping such that if no child exceeds a node's threshold, we return that parent label. This allows predictions to degrade gracefully on uncertain or OOD cases. More details are provided in the Appendix.

We evaluate HSMs using top-1 accuracy, hierarchical distance, DCA distance, and prediction depth. Top-1 accuracy is the fraction of correct diagnosis-level predictions in the cohort (OOD labels are mapped to their deepest ancestor in the ID hierarchy). Hierarchical distance is the number of edges between prediction and ground truth in the WHO5 tree. The DCA is the lowest shared ancestor of prediction and truth; we report the number of edges from the DCA to the ground-truth label.

### 2.5.2 DEEPHIT: SURVIVAL PREDICTION

DeepHit, a neural network for survival analysis, leverages LATTICEs to model time-to-event risk for patient outcome prediction Lee et al. (2018). Similar to the HSM, LATTICEs are concatenated to their paired clinical embeddings before being passed through the MLP backbones of two parallel DeepHit networks, one modeling overall survival (OS) and the other progression-free survival (PFS) (Figure 2d). Each DeepHit model outputs a $T$-dimensional vector containing risk scores for each time interval. We evaluate performance using concordance-index (c-index).

## 3 RESULTS

### 3.1 BLOOM-HiCLIP LEARNS TAXONOMY-ALIGNED LATTICEs

We first evaluated whether BLOOM-HiCLIP effectively reconstructs both DNA and RNA modalities to quantify missing-modality imputation quality. We observe high cosine similarity between the imputed and true target embeddings using Top-1 and Oracle@5 neighbor retrieval across all evaluation splits, indicating LATTICEs places each patient near biologically similar samples whose target-modality representations yield accurate imputations even when self-retrieval is disallowed (Figure 3a). Reconstruction performance is statistically indistinguishable between RNA → DNA and DNA → RNA, suggesting both towers support accurate bidirectional imputation.

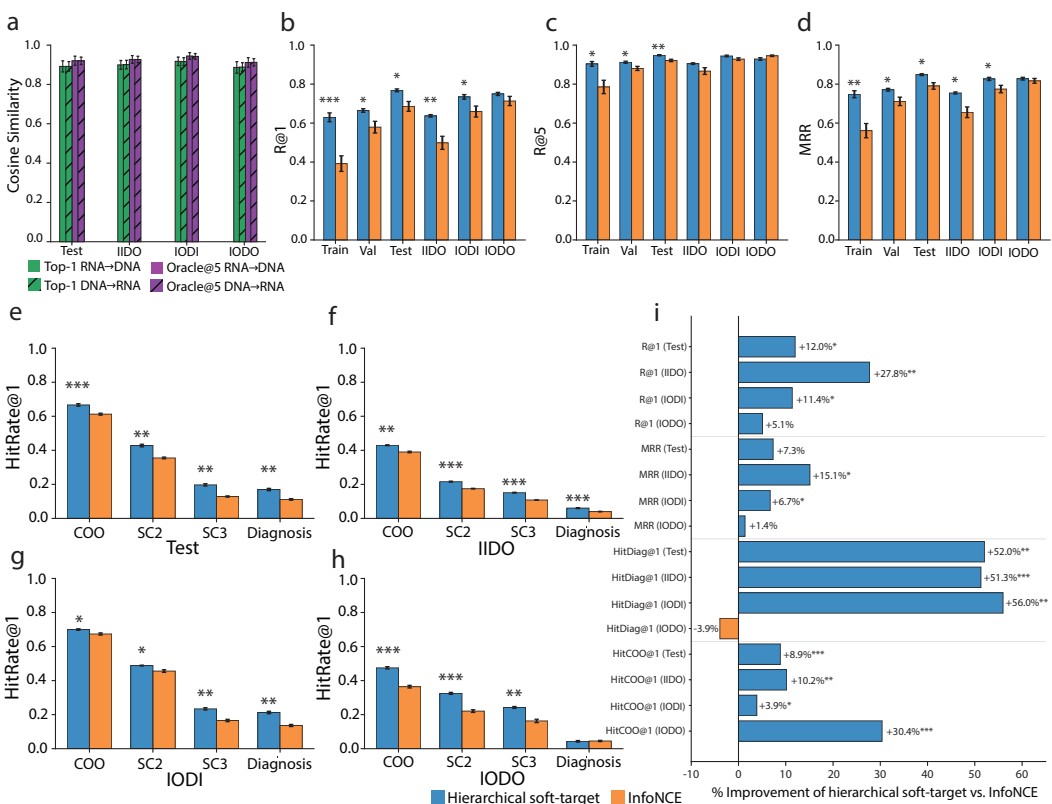

Figure 3: (a) Imputation quality, (b–d) retrieval, (e-h) taxonomic recall, and (i) performance summary of BLOOM-HiCLIP. * indicates p<0.05, ** indicates p<0.01, *** indicates p<1e-3.

BLOOM-HiCLIP achieved higher paired-sample retrieval than the InfoNCE ablation despite In-foNCE directly optimizing paired recall (Figures 3b–d). BLOOM-HiCLIP outperformed InfoNCE in R@1 and MRR on the train, validation, test, IIDO, and IODI sets, and in R@5 on the train, validation, and test sets (p<0.05). While gains on IODO are not significant, BLOOM-HiCLIP maintained strong performance (R@1 = 74.9±2.0%, R@5 = 92.9±1.6%, MRR = 0.828±0.00).

Finally, we tested whether BLOOM-HiCLIP learns taxonomy-consistent structure using HitRate@1 across WHO5 (Figures 3e-h). Our hierarchical soft-target loss significantly (p< 0.05) outperformed

InfoNCE across all taxonomic levels in the test, IIDO, IODI, and IODO sets except at the diagnosis level on IODO. These results indicate LATTICEs preserve taxonomic signal despite this being secondary to paired retrieval in the loss function.

Overall, BLOOM-HiCLIP achieved broadly higher performance across retrieval and taxonomic-alignment metrics compared to the InfoNCE-trained ablation model, demonstrating that incorporating hierarchy-aware supervision improves both paired cross-modal matching and biologically meaningful neighborhood structure without sacrificing generalization to held-out and out-of-distribution cohorts or imputation quality (Figure 3i).

## 3.2 LATTICEs TRANSFER TO HIERARCHICAL DIAGNOSIS & SURVIVAL TARGET TASKS

LATTICE-trained HSM models achieve strong discrimination across the hierarchy with high validation set micro-averaged AUROC at all internal nodes (e.g., 98.1% AUROC for COO) (Figure 4a). The HSM significantly outperforms all baselines in hierarchical distance, DCA distance, and top-1 accuracy ($p < 0.05$) even on OOD institutions and diagnoses (Figures 4b-c). Our respective test set hierarchical and DCA distances of 2.0±0.0 and 1.6±0.1 paired with observed prediction depth indicate errors are mostly local, likely due to early stopping on the right taxonomic paths, not wrong-branch jumps. Test set prediction depth of 2.8±0.2 paired with top-1 accuracy of 34.0±3.5% indicate our HSM makes fairly deep subtype-level calls, with roughly one-third of total predictions being exactly correct at the granular subtype level, consistent with frequent near-miss errors or early stopping within the correct lineage. The LATTICE-supplemented HSM achieves the best performance even on OOD institutions and diagnoses with a top-1 accuracy of 15.5±4.0% on the IODO set. Overall, this indicates LATTICEs remain informative even when patients come from new hospitals or rare entities, supporting the clinical utility of calibrated hierarchical prediction when exact subtypes are ambiguous or rare.

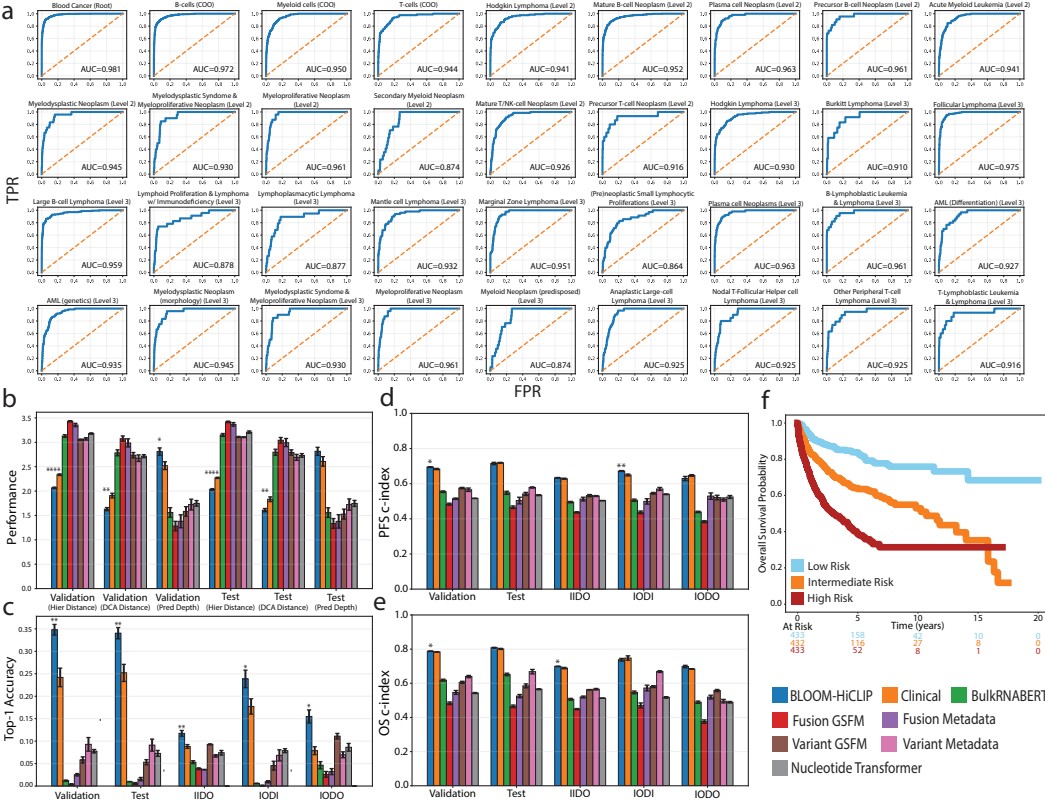

Figure 4: (a) HSM micro-averaged ROC curves at WHO5-hierarchy internal nodes. (b-c) HSM and (d-e) DeepHit performance vs. baselines. (f) KM curve of predicted risk tertiles on IIDO set.

LATTICE-trained DeepHit models show strong survival stratification across evaluation sets, achieving validation c-indices of 0.695±0.01 and 0.788±0.00 respectively (Figures 4d-e). Performance remains robust under institution and diagnosis shift with respective IODO PFS and OS c-indices of 0.630±0.03 and 0.698±0.02, highlighting the generalizability of prognostic signal learned from LATTICE and clinical covariates. Compared to baselines, LATTICE-enhanced DeepHit models significantly outperform models trained on genomic inputs alone and match or exceed clinical-only models. Consistent with these results, Kaplan–Meier (KM) curves of 3-year OS predicted risk groups on IIDO show clear separation despite diagnosis shift (p=3.47e-15) (Figure 4f).

## 4  DISCUSSION

BLOOM-HiCLIP is the first hierarchical multi-omic CLIP framework for blood cancers, designed to address two fundamental challenges in clinical multi-omics: entire-modality missingness and distribution shift across institutions and rare diagnoses. Trained on the largest multi-omic blood cancer cohort to date (ABCG: 8,281 tumors; 165 diagnoses; 28 institutions), BLOOM-HiCLIP aligns RNA and DNA-derived patient views into a shared embedding space that supports cross-modal retrieval, missing-modality imputation, and downstream transfer to clinically relevant tasks.

A central finding is that hierarchy-aware soft-target loss improves both paired retrieval and taxonomy-aligned neighborhood structure compared to InfoNCE. This suggests WHO5 reflects biological structure in multi-omic profiles such that patients with related diagnoses share features that flat contrastive objectives fail to exploit. The benefit is particularly pronounced under OOD institutions and diagnoses where BLOOM-HiCLIP maintains higher retrieval and hit rates across WHO5 levels. These results indicate that encoding clinical ontology into the training objective yields representations that generalize more reliably than instance-level discrimination alone.

BLOOM-HiCLIP achieves these gains while remaining parameter-efficient. By freezing foundation encoders and training only lightweight projection heads, transformers, and scalar parameters, we obtain strong performance with 33.4x fewer trainable parameters than end-to-end fine-tuning would require. While we do not claim this approach outperforms or matches full fine-tuning, our results demonstrate that rich multi-omic representations can be learned without the computational cost of updating billion-parameter backbones, an important consideration when scaling to larger cohorts.

The learned embeddings transfer effectively to downstream clinical tasks. For hierarchical diagnosis, the LATTICE-trained HSM achieves strong discrimination across WHO5 levels, with prediction errors that are predominantly local rather than wrong-branch jumps, consistent with the intended back-off behavior under uncertainty. For survival prediction, DeepHit models combining embeddings with clinical covariates achieve robust risk stratification even on OOD cohorts. Notably, the performance gain over baselines is larger for diagnosis than prognosis, likely reflecting both the hierarchical training objective and the stronger contribution of clinical covariates to survival outcomes.

Several limitations warrant acknowledgment. First, while BLOOM-HiCLIP outperforms single-modality baselines, we did not compare against established multi-omic integration methods such as MOFA+ (Argelaguet et al., 2020). Such comparisons would further contextualize our contribution. Second, top-1 diagnostic accuracy remains modest in absolute terms, reflecting the inherent difficulty of fine-grained classification across dozens of subtypes with limited training examples. Finally, survival prediction improvements over clinical-only models were incremental, suggesting genomic embeddings may complement rather than replace traditional clinical prognostic factors.

Several directions could extend this work. Incorporating additional modalities such as copy-number alterations, viral detection, or histopathological images may improve performance and reduce brittleness when any single modality is missing. Fine-tuning the frozen foundation encoders could quantify the accuracy-compute trade-off and test whether blood-cancer-specific adaptation yields further gains. Beyond model improvements, interpretability methods that identify which molecular events drive patient similarity could help clinicians understand and trust imputed representations.

In summary, BLOOM-HiCLIP demonstrates that hierarchy-aware contrastive alignment can transform incomplete, heterogeneous multi-omic data into representations that support imputation and transfer robustly to relevant target tasks. By respecting disease taxonomy and efficiently leveraging biological foundation models, this framework provides a scalable foundation for multi-modal precision hematologic oncology.

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

# A    APPENDIX

## A.1    BLOOD CANCER TAXONOMY

The ABCG cohort's contains 4 main data modalities: genetic variants, bulk gene expression, fusion calling, and clinical data. Mutation profiles show recurrent alterations in known oncogenic genes and variant distribution consistent with the literature such as the overrepresentation of *TET2* alterations in T/NK-cell cancers (p=1.86e-10) (Figure S1a) Carty (2023). t-SNE visualization of bulk transcriptomic profiles indicates the WHO5 structure is represented in raw genetic inputs, revealing clear separation between myeloid and lymphoid (B and T/NK-cell) malignancies as well as isolated clustering within lineages (Figure S1b) van der Maaten & Hinton (2008). Fusion analysis reveals extensive heterogeneity across and within these COO lineages while confirming known tumor biology (Figure S1c). For instance, we observe *PML-RARA* fusions are significantly enriched in Acute Promyelocytic Leukemia (APL) cases, a myeloid malignancy characterized by that fusion (p=3.45e-75) Ryan (2018). Our clinical data underscores the diversity of our cohort. Among patients with reported sex, race, and age, 42.4% were female, 23.0% were non-white, and the median age at diagnosis was 59.9 years. PFS and OS were used to measure clinical outcomes. The median survival time was 3.7 years (95% CI 3.4-4.1) with 2,594 (55.1%) events and 15.8 years (95% CI 15.0-17.8) with 1,671 (28.8%) events for PFS and OS respectively. KM curves reveal notable survival stratification across COOs, with B-cell malignancies exhibiting better PFS (p=3.10e-54) and OS (p=3.63e-93) than malignancies from other COOs (Figures S1d-e).

## A.2    SEQUENCING

Our study uses cases from the ABCG dataset, an international collaborative effort from 28 clinical sites in which we aimed to collect and sequence cases that represent the full blood cancer family. Formalin-fixed, paraffin-embedded tumor biopsies were collected in accordance with an IRB-approved protocol. All tissue biopsies were deidentified and diagnoses were confirmed by expert panel-based pathology review using purpose-built tools. Each tumor biopsy has gene expression and fusion calls from whole transcriptome sequencing as well as matched gene mutation data from whole exome and targeted panel sequencing. All nucleic acid extractions and sequencing library preparations were performed with Duoseq (Research Kit EPXv3, Data Driven Bioscience, Durham, NC) following the manufacturer's protocol. All libraries were sequenced on the Illumina platform according to manufacturer recommendations.

### A.2.1    DNA SEQUENCING ANALYSIS

FASTQ files containing DNA sequencing reads were trimmed using Trimmomatic (v0.39) in paired end mode to remove adapter sequences and low-quality reads Cock et al. (2010); Bolger et al. (2014). Using Sentieon BWAmem (v201911) with the default settings, DNA reads were aligned to the human genome (GRCh38.p12, with a PAR mask on chrY) Freed et al. (2017). PCR duplicate reads were marked using Picard (v2.8.1) Project. Picard, FASTQC (v0.11.8), and samtools (v1.13) were used to extract quality control metrics Li et al. (2009).

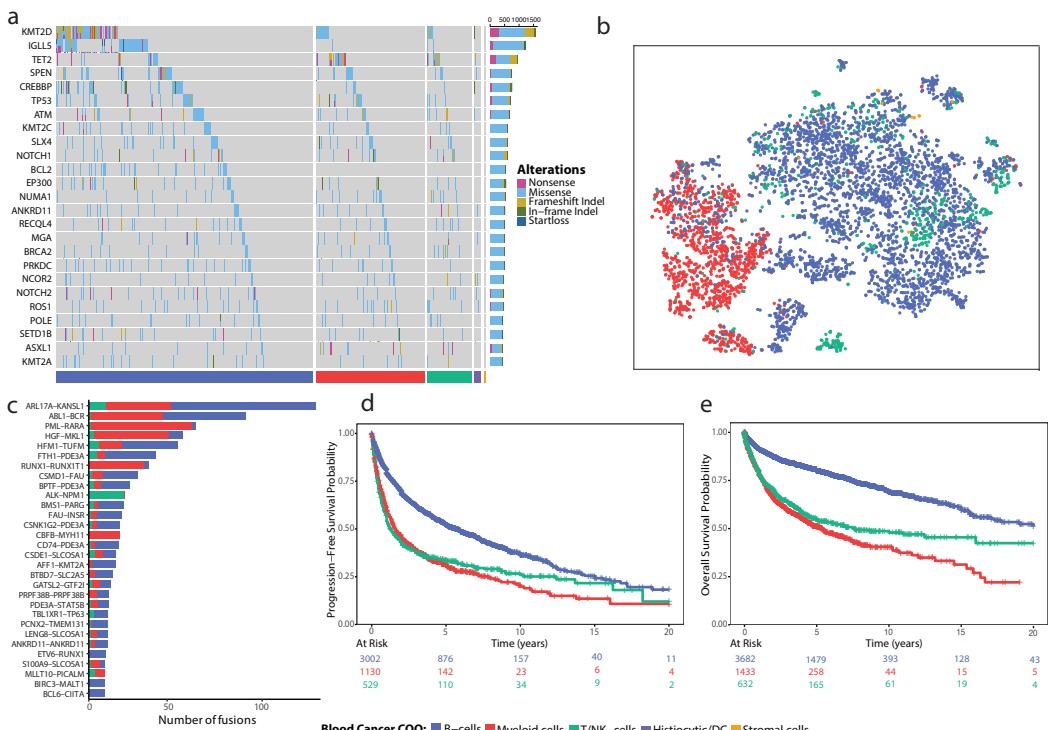

Figure S1: (a) Oncoprint of the 25 most recurrently mutated genes in the cohort column-split by COO. (b) Gene expression t-SNE colored by COO. (c) Most common fusions colored by COO. (d) PFS and (e) OS KM curves stratified by COO. Histiocytic/dendritic and stromal cell malignancies were excluded in d-e as few such cases had survival data.

### A.2.2 VARIANT CALLING ANALYSIS

Variant calling analysis was performed using DNA reads from exome samples and targeted panel sequencing where available. Variants were called using a pipeline that uses the union of variants called by each of Strelka2 (v2.9.10), DeepVariant (v1.1.0), and Sentieon Haplotyper (v201911) Saunders et al. (2012); Poplin et al. (2018); Freed et al. (2017). Synonymous single nucleotide variants (SNVs), variants found in the separate panel of normal control samples, and variants with a population frequency of greater than 0.01 reported in the gnomAD Karczewski et al. (2020) databases were excluded. We only included variants included in the union of genes previously described by three major blood cancer genomics profiling studies Chapuy et al. (2018); Reddy et al. (2017); Wright et al. (2020).

For each variant, we retain a rich set of per-event annotations from the Sentieon-based calling pipeline, including genomic locus, reference and alternate alleles, variant type (e.g., SNV/indel), and quality evidence (e.g., depth and allele support) together with downstream functional metadata such as ClinVar clinical significance and associated disease names. Additionally, we have standardized deleteriousness scores such as CADD that summarize predicted functional impact. To provide sequence context for the DNA encoder, we further extract a fixed-length window from the hg38 reference FASTA centered on each event—specifically, a 2,049 bp segment consisting of 1,024 bp upstream and 1,024 bp downstream of the variant site (with the variant centered within the window) which is then tokenized as nucleotide input by Nucleotide Transformer.

At the patient level, event-wise variant metadata are pooled into a single representation by aggregating across all variants observed in that individual. This pooled patient-level metadata vector is concatenated with DNA-seq quality-control features (e.g., coverage metrics) to form the final DNA-side input to BLOOM-HiCLIP, ensuring that downstream modeling jointly reflects both biological variant content and assay quality. Any missing features were marked with a missingness indicator and, for continuous features, were imputed using the train set's mean.

### A.2.3 RNA SEQUENCING & GENE EXPRESSION ANALYSIS

FASTQ files containing RNA-Seq reads were trimmed using Trimmomatic (v0.39) in paired end mode to remove Illumina specific adapter sequences and low-quality reads Bolger et al. (2014). STAR aligner (v2.7.1) was utilized for mapping RNA reads to the human genome and transcriptome (GRCh38.p12, with a PAR mask on chrY) Dobin et al. (2013). Reads that had at least one primary alignment to the human genome were extracted using samtools (v1.13) and additionally filtered based on manufacturer recommendations using GATK (v4.1.2.0) Li et al. (2009); McKenna et al. (2010). Transcript quantification was performed using salmon (v1.2.1), and the transcript-level data were summarized to the gene level using the tximport (v1.14.2) library in R Patro et al. (2017); Soneson et al. (2015). The result was a matrix representing raw read counts per gene per patient sample. This matrix was restricted to include only the expression of the protein-coding genes captured by exome sequencing. Genes were transcripts per million (TPM) normalized in accordance with the desired input for BulkRNABERT.

### A.2.4 FUSION ANALYSIS

Fusions were identified using Arriba (v2.1.0) Uhrig et al. (2021). We retain per-event metadata such as the partner genes, breakpoint coordinates, read-level support, and caller-specific confidence/filters, along with annotations describing coding impact (e.g., in-frame status and predicted protein consequence) and recurrent/known fusion labels when available. Arriba also outputs the predicted peptide sequence of the fusion protein for each event and embed this sequence with ESM-2, using it as a protein-level representation of the fusion's functional content.

Fusion events are then pooled into a patient-level representation by aggregating across all fusions observed in an individual (e.g., combining summary statistics reflecting fusion burden and confidence-weighted evidence). This pooled patient-level fusion metadata is concatenated with RNA-seq QC metrics (e.g., RNA to DNA ratio and coverage/expression quality indicators) to form the patient-level fusion metadata representation.

## A.3 FOUNDATION ENCODERS

BLOOM-HiCLIP leverages 5 pretrained foundation encoders—BulkRNABert (InstaDeep, a), Nucleotide Transformer v2 (500M, multi-species) (InstaDeep, b), S-PubMedBERT-MS-MARCO (Deka), ESM-2 t33 650M UR50D (Meta AI), and GSFM (Ma'ayan Lab). Table S1 reports the modality, embedding dimension, and trainable parameters of each foundation encoder.

Table S1: Foundation encoders used in BLOOM-HiCLIP.

| Model | Modality | Embedding dim | Parameters |
|---|---|---|---|
| ESM-2 | Protein (fusion peptide) | 1280 | 651,040,661 |
| Nucleotide Transformer | DNA (nucleotide sequence) | 1024 | 498,345,436 |
| PubMedBERT | Text (ClinVar disease terms) | 768 | 109,482,240 |
| GSFM | Gene / gene-set embeddings | 256 | 15,116,232 |
| BulkRNABERT | Bulk RNA expression | 256 | 6,006,130 |

## A.4 BLOOM-HiCLIP DEVELOPMENT

Below is a detailed description of each tower in BLOOM-HiCLIP.

### A.4.1 RNA TOWER TOKENIZATION

The RNA tower produces an embedding $z_i^{\mathrm{RNA}} \in \mathbb{R}^d$ for patient $i$ with $F$ fusions from the following embeddings projected to a $d$-dimensional representation (Figure 2a):

1. **Bulk expression token:** BulkRNABERT patient embedding
2. **Fusion patient tabular token:** Patient-level vector of RNA-seq QC and fusion metadata
3. **Fusion gene-set token:** If $F \geq 1$, a GSFM embedding of all unique genes involved in fusions. Otherwise, a learned <NO_CALLED_FUSIONS> token

4. **Fusion event tokens:** Each fusion event is represented by a `<FUSION_EVENT>` token built from the following components concatenated and projected to $d$:

   (a) Gene-pair embedding formed from projections of GSFM embeddings of each fusion partner gene and then fusing them with another projection

   (b) Projected ESM-2 embedding of the fusion protein peptide sequence when available. Otherwise, a learned `<MISSING_PEPTIDE_SEQ>` token

   (c) Projected tabular fusion metadata vector

The top $K = 32$ events by coverage per patient are kept. The RNA token sequence therefore contains $F + 3 \leq 35$ $d$-dimensional tokens where the 3 tokens always included are the projected BulkRNABERT embedding, the patient-level tabular embedding, and the GSFM gene set embedding or `<NO_CALLED_FUSIONS>` if $F = 0$.

### A.4.2 DNA TOWER TOKENIZATION

The DNA tower produces an embedding $z_i^{\mathrm{DNA}} \in \mathbb{R}^d$ for patient $i$ with $V$ variants from the following embeddings projected to a $d$-dimensional representation (Figure 2b):

1. **Variant patient tabular token:** Patient-level vector of DNA-seq QC and variant metadata

2. **Variant gene-set token:** If $V \geq 1$, a GSFM embedding of all unique genes involved in variants. Otherwise, a learned `<NO_CALLED_VARIANTS>` token

3. **ClinVar patient token:** ClinVar-specific patient token from the following:

   (a) ClinVar patient-level tabular vector if a patient has any ClinVar-associated variants. Otherwise, a learned `<NO_CLINVAR>` token

   (b) Attention-pooled PubMedBERT embeddings for each unique ClinVar disease name involved in variants for the patient. If the patient contains ClinVar variants, but no disease names are present, we output a learned `<NO_CLNDN>` token

4. **Variant event tokens:** Each variant event is represented by a `<VARIANT_EVENT>` token built from the following components concatenated and projected to $d$:

   (a) Projected Nucleotide Transformer embedding

   (b) Projected non-ClinVar tabular variant metadata vector

   (c) Projected GSFM gene embedding for the mutated gene

   (d) Projected ClinVar tabular metadata vector if the variant is in ClinVar. Otherwise, a learned `<VAR_NOT_IN_CLINVAR>` token

The top $K = 32$ events by coverage per patient are kept. The DNA token sequence therefore contains $V + 3 \leq 35$ $d$-dimensional tokens where the 3 tokens always included are the projected patient-level tabular embedding, the GSFM gene set embedding or `<NO_CALLED_VARIANTS>` if $V = 0$, and the ClinVar embedding or `<NO_CLINVAR>`.

### A.4.3 BLOOM-HICLIP TRAINING PROTOCOL

For CLIP training, we used the subset of IIDI cases with both RNA and DNA modalities available (n=3,715). We then performed a random 90/10 train/test split and then a separate 80/20 random train/validation split on the resulting train set both stratified by diagnosis. The resulting train and validation sets were used for model training, hyperparameter tuning, and early stopping. The test and RNA-DNA paired subsets of the IIDO, IODI, and IODO sets were used exclusively for post-training evaluation.

### A.4.4 BLOOM-HICLIP ARCHITECTURE & HYPERPARAMETER SELECTION

In Table S2 we present the architecture of the BLOOM Hi-CLIP model as well as hyperparameters selected from our BO protocol. We train BLOOM-HiCLIP using AdamW with separate cosine-scheduled learning rates for the token-transformer, the projection and pooling parameters, the temperature parameter, and the decay and same patient-weight parameters. Weight decay is applied to the transformer and projection parameters. We tune hyperparameters using Bayesian Optimization

(BO) with `Optuna` to maximize mean cross-modal retrieval performance on the validation set, using Recall@5 as the objective Akiba et al. (2019). The search space includes: $d$, learning rates, weight decay, batch size, number of transformer heads, transformer depth, feedforward hidden dimension, and depths and widths for the metadata, gene-set pair, and event-level token projections. We perform 50 BO trials with a patience of 3 based on validation set R@5.

**BLOOM-HiCLIP evaluation metrics** We evaluate cross-modal retrieval using cosine-similarity matrices between L2-normalized query and target embeddings, where the true match for query $i$ is the paired patient on the diagonal from which we report Recall@k (R@k), mean/median rank, and mean reciprocal rank (MRR). To measure taxonomy-consistent neighborhood structure, we compute HitRate@1 at each WHO5 level (diagnosis, supercategory 3, supercategory 2, and COO), counting a hit if, excluding the query patient, the top retrieved neighbor shares the same label. For missing-modality imputation, we mask self-retrieval and reconstruct the target-modality embedding from retrieved neighbors, reporting Top-1 cosine similarity (cosine between the true target embedding and the embedding of the single best retrieved neighbor) and Oracle@5 cosine similarity (the highest such similarity among the top-5 retrieved candidates) for both RNA→DNA and DNA→RNA retrieval directions.

Table S2: Selected BLOOM-HiCLIP hyperparameters.

| Category | Value |
|---|---|
| *Model / architecture* | |
| $d$ | 512 |
| Transformer heads ($n_{\text{head}}$) | 4 |
| Transformer layers ($L$) | 2 |
| FFN width (ff_mult) | 3072 |
| Dropout | 0.183 |
| | |
| *Projection heads* | |
| Event token projector depth | 2 |
| Event token projector output dims | [1536, 2048] |
| Gene pair projector depth | 0 (none) |
| Gene pair projector output dims | [] |
| Metadata projector depth | 2 |
| Metadata projector output dims | [1024, 512] |
| Foundation encoder projector depth | 1 |
| Foundation encoder output dims | [512] |
| | |
| *Optimization* | |
| Batch size | 32 |
| Transformer LR | $1.09 \times 10^{-4}$ |
| Projection LR | $1.09 \times 10^{-4}$ |
| Temperature ($\tau$) LR | $5.43 \times 10^{-4}$ |
| Same patient weight & decay ($\alpha$) LR | $1.09 \times 10^{-3}$ |
| Weight decay | 0.086 |
| Gradient clipping | 1.0 |
| | |
| *Training / early stopping* | |
| Initial temperature ($\tau_0$) | 0.07 |
| Initial decay ($\alpha_0$) | 0.768 |
| Initial same patient weight | 10 |
| Max epochs | 25 |
| Validation Recall@k used for selection | $k = 5$ |

## A.5 TARGET MODEL DEVELOPMENT

### A.5.1 TARGET MODEL TRAINING PROTOCOL

To prevent leakage during target model training, we performed the same train/validation/test diagnosis-stratified split on IIDI samples with either missing DNA or RNA modalities and con-

catenated the resulting sample sets to the corresponding train/validation/test sets used in CLIP training. This ensures the target model train set is supplemented by cases with imputed data while still containing all samples in the set used to train BLOOM-HiCLIP. Missing continuous features were imputed using the train set mean. All missing clinical features were marked with an indicator. Target model architectures are provided in Tables S3-4. Baselines follow the same training protocol and splits, but use single modalities in isolation with missing modalities imputed by mean-pooling train-set embeddings and appending missingness indicators.

### A.5.2 HIERARCHICAL SOFTMAX

**Loss function**   Because the WHO-defined taxonomy is a rooted DAG, there exists a unique path from the root to each leaf. At each internal node of this hierarchy, the HSM has a linear layer with a softmax activation function that outputs the conditional probability distribution across its child nodes. During inference, the model computes the joint probability of each diagnosis as the product of conditional probabilities along the corresponding taxonomic path, enabling consistent predictions that respect the blood cancer ontology.

Given this nested prediction task reliant on conditional probabilities, we implemented a loss function that balances confidence along the true diagnostic path with calibrated behavior at off-path nodes (Equation 1) Ridnik et al. (2021); Linderman et al. (2022). The loss includes a path-specific cross-entropy term that encourages high confidence along the ground truth hierarchical path (Equation 2). It also weighs prediction errors at internal nodes with many descendant nodes more than errors at internal nodes with fewer descendants as higher-level misclassifications lead to a greater number of downstream predictions being incorrect (Equation 3). In addition, for all internal nodes not on the sample's path, we apply a term that penalizes deviations from a uniform distribution over those nodes' children (Equation 4). This prevents the HSM from becoming overconfident in branches outside the true prediction path, improving calibration and robustness, which is particularly important when handling samples from OOD subtypes.

$$L = L_{soft} + \alpha L_{other} \tag{1}$$

$$L_{soft} = \sum_{n \in Z} W_n \sum_{(x,y) \in D_n} H[\text{onehot}_n(y) || p_n(x)] \tag{2}$$

$$W_n = \frac{|\{j \in \{1, ..., N\} : n \in anc(j)\}|}{|Z|} \tag{3}$$

$$L_{other} = \sum_{n \in Z} \sum_{(x,y) \in D_{\neg n}} H[U(|ch(n)|) || p_n(x)] \tag{4}$$

where $n$ is a node in the set $Z$ of all internal nodes with ancestors $anc(n)$ and children $ch(n)$. $D_n$ is the set of tuples $(x_i, y_i)$ of samples whose ancestors contain $n$ in which $x_i$ represents sample $i$'s input features and $y_i$ is its ground-truth leaf node label. $H[s||t]$ is the cross-entropy from $s$ to $t$. $p_n(x_i)$ is the model's predicted probability distribution over $ch(n)$ for sample $i$. $\alpha$ is a hyperparameter dictating the relative weight of $L_{soft}$ to $L_{other}$ in the cumulative loss function $L$.

**Adaptive confidence thresholding**   To mitigate overconfident misclassifications, we evaluated a node-specific confidence threshold parameter $\tau$. We first computed node-level confidence scores by evaluating the HSM's predicted path probabilities along all possible root-to-leaf paths and recording, for each node, the probability of selecting its correct child (Equation 5). For each $n$, this produced a set of scores $\{s_i(n, c), y_i\}_{i=1}^{N_n} \forall c \in ch(n)$ where $s_i$ is sample $i$'s cumulative predicted path probability from the root to one of $n$'s children $c$ and $y_i \in \{0, 1\}$ indicates whether the child lies along the true path for the set of samples $N_n$ at node n.

$$s_i(n, c) = (\prod_{j=1}^{l-1} P(v_{j+1} | v_j, x_i)) P(c | n, x_i) \tag{5}$$

where the path to $n$ is $\gamma(n) = (v_{1=root}, v_2, ..., v_n)$. We then used these node-wise data to estimate the threshold $\tau_n$ that ensures minimal desired precision $\tau \in [0, 1]$ at $n$ (Equations 6-7).

$$Precision_n(t) = \sum_{c \in ch(n)} \sum_{i \in N_n} \frac{\mathbf{1}(s_i(n, c) \geq t)y_i}{\mathbf{1}(s_i(n, c) \geq t)} \tag{6}$$

$$\tau_n = min\{t|Precision_n(t) \geq \tau\} \tag{7}$$

During inference, the classifier traverses the taxonomy top-down. Starting at the root, it selects the child (COO) with the highest conditional probability and continues along that path greedily selecting the most probable child at each internal node, updating the cumulative path probability. If the cumulative probability at a given $n$'s prediction over $ch(n)$ along this path falls below $\tau_n$, the model halts prediction and returns $n$, the last confidently assigned taxonomic class. This adaptive thresholding causes the HSM to produce shallower, higher-level classifications for ambiguous or OOD samples while maintaining high precision in confident cases.

**Hyperparameter tuning**  We optimized HSM architectures and hyperparameters using 100 BO trials with a patience of 5 on the validation set HSM loss (Equation 1) (Table S3). We optimized the number and dimensions of hidden layers as well as output dimension and dropout for both the shared MLP head and clinical data MLP encoder. We also optimized hyperparameters included batch size, AdamW learning rate, weight decay, and the loss function $\alpha$ parameter. Post-training, we computed conditional probabilities on the validation set, converted them to cumulative path probabilities (Equation 5), calibrated node-wise stopping thresholds by sweeping a grid of minimum precision thresholds (Equations 6-7). At each precision threshold, we compute hierarchical distance select the threshold that minimizes hierarchical distance. This hierarchical distance is the BO study minimization objective metric. We utilize the same BO protocol for the HSM baselines.

Table S3: Selected HSM hyperparameters.

| Category | Value |
|---|---|
| *Shared encoder* | |
| Number of layers | 2 |
| Hidden layer dims | [2048, 512] |
| Dropout | 0.10 |
| *Clinical encoder* | |
| Number of layers | 1 |
| Hidden layer dims | [64] |
| Output dimension | 256 |
| Dropout | 0.50 |
| *Optimization* | |
| Batch size | 128 |
| LR | $8.13 \times 10^{-4}$ |
| Weight decay | $1.50 \times 10^{-4}$ |
| Hierarchy loss scale ($\alpha$) | 0.0274 |
| Precision threshold ($\tau_{min}$) | 0.85 |
| Max epochs | 100 |

### A.5.3   DEEPHIT

DeepHit is a neural-network–based survival model that directly learns the joint distribution of event occurrence and time through discrete-time likelihood estimation. In our implementation, we used `DeepHitSingle` from the `pycox` library to model PFS and OS Lee et al. (2018); Kvamme (2019). Our DeepHit models output a discrete probability mass function over $T$ pre-specified time intervals (0 to 20 years with a step size of 1 month) corresponding to the conditional event-time distribution $P(T = t|X)$.

**Hyperparameter tuning**   We performed 100 trials of BO to identify optimal architectures and hyperparameters for both DeepHit models (Table S4). We optimized the following hyperparameters: AdamW learning rate, weight decay, and batch size. We also tuned the number and dimensions of hidden layers and dropouts of the shared MLP backbones and clinical data MLP encoders. We used default DeepHit loss on the validation set for the training early stopping with a patience of 5. Post-training we use validation set c-index as the BO objective maximization metric.

Table S4: Hyperparameters for DeepHit models.

| **DeepHit OS** | | **DeepHit PFS** | |
|---|---|---|---|
| **Component** | **Value** | **Component** | **Value** |
| *Clinical encoder* | | *Clinical encoder* | |
| Number of layers | 4 | Number of layers | 1 |
| Hidden layer dims | [512, 1024, 16, 256] | Hidden layer dims | [256] |
| Output dim | 128 | Output dim | 256 |
| Dropout | 0.45 | Dropout | 0.25 |
| *Shared encoder* | | *DeepHit head (MLP)* | |
| Number of layers | 1 | Number of layers | 3 |
| Hidden layer dims | [64] | Hidden layer dims | [128, 256, 64] |
| Dropout | 0.40 | Dropout | 0.35 |
| *Training* | | *Training* | |
| LR | $1.64 \times 10^{-4}$ | LR | $5.30 \times 10^{-3}$ |
| Weight decay | $1.05 \times 10^{-6}$ | Weight decay | $2.48 \times 10^{-4}$ |
| Batch size | 256 | Batch size | 32 |
| Max epochs | 1000 | Max epochs | 1000 |

## A.6   COMPUTE RESOURCES

For CLIP model development, BO studies were run on Google Colab using an NVIDIA A100 GPU. For HSM model development, BO studies were run on Google Colab using an NVIDIA L4 GPU. For DeepHit model development, BO studies were run on Google Colab using CPU.

## A.7   SIGNIFICANCE TESTS

Categorical association test significance values were evaluated using 2-tailed Fisher's exact tests. Significance of KM curve stratification was evaluated using 2-tailed multivariate log-rank significance tests. 95% confidence intervals comparing model performance were calculated using a t-distribution.

## A.8   DATA VISUALIZATION

t-SNE plots were generated using the `TSNE` method from `sklearn` with perplexity parameters of 50 Pedregosa et al. (2011). KM curves were generated using the `KaplanMeierFitter` and `add_at_risk_counts` methods from the `lifelines` library Davidson-Pilon (2019). Plots were generated using the `matplotlib` and `seaborn` Python libraries except for the sunburst plot and oncoprint which were generated with the `plotly` Python library and the `ComplexHeatmap` R package's `oncoPrint` function respectively Hunter (2007); Waskom (2021); Inc. (2015); Gu et al. (2016).

## A.9   LLM USAGE DISCLOSURE

ChatGPT-5.2 was used for proofreading this manuscript and reviewing code.

## A.10   MEANINGFULNESS STATEMENT

BLOOM-HiCLIP learns meaningful representations of life by aligning two complementary views of the same tumor into a shared embedding space: gene expression and fusion events in RNA space and

genomic alternations in DNA space. By training these representations to respect clinically grounded hematologic taxonomies, the model encodes relatedness in a geometric structure, enabling robust retrieval, transfer, and interpretation across rare and OOD diagnoses. The shared space supports missing-modality imputation, letting us study tumors with incomplete assays without discarding patients. This yields representations that reflect clinically-defined taxonomic organization that improve downstream diagnosis and prognosis.

### A.10.1 ACKNOWLEDGMENTS

We thank the current and past members of the Sandeep Dave Lab, especially Ayush Batra, Fadzai Chinyengetere, Tushar Dave, Lanie Happ, Rachel Kositsky, Cassandra Love, Dennis Owusu, Razvan Panea, Jessi Rodgers, Veronica Russell, Jennifer Shingleton, Devang Thakkar, and Shari Tian for their work in sample processing and bioinformatic pipeline development. We also thank all collaborators in the ABCG consortium for providing and clinically reviewing samples; Katherine Dura of the Bill Majoros Lab for her contribution to the variant detection pipeline; and Nathaniel Blalock, Coban Brooks, Benjamin Perry, and Srinath Seshadri of the Philip Romero Lab for their advice in model development.

