# OpenReview forum: "Hierarchical Multi-Omic CLIP for Missing-Modality Imputation & Transfer Learning in Blood Cancers"
_ICLR.cc/2026/Workshop/LMRL — ICLR 2026 Workshop LMRL Poster_

### Official Review · Reviewer_5Ruq · 2026-02-20
**Novel Foundation Model for Blood Cancer with Evaluation and Risk Stratification Limitations**

**Rating:** 7
**Confidence:** 3

**Review:**

Summary:
The study introduces BLOOM-HiCLIP a multi-omics foundation model for blood cancer. BLOOM-HiCLIP introduces a novel architecture which combined multiple existing single-modality foundation models trained through a custom CLIP-adapted contrastive loss. The evaluation covers latent space retrieval, hierarchical disease typing and patient risk stratification in in- and out-of-distribution settings.

Pro:
- The study introduces an interesting and novel problem setting of building a multi-omics foundation model for blood cancer.
- The study proposed a novel supervised contrastive training loss well tailored to the hierarchical taxonomy of blood cancer. In an ablation, this loss shows performance gains over a more simple InfoNCE loss.
- The evaluation protocol is exemplary in its treatment of potential batch effects, including in-distribution and out-of-distribution analyses stratified by both unseen institutions and previously unseen disease subtypes.
- Hierarchical softmax models trained on BLOOM-HiCLIP embeddings achieve good performance on disesase prediction improving over single-modality baselines.

Contra:
- For survival stratification (Fig. 4d-e) (arguably the most important task) the performance improvement compared to using only clinical data is marginal at best.
- Term "missing modality imputation" frequently used throughout the manuscript seems inappropriate. This expression suggests that the model would be able to impute/reconstruct a missing modality, e.g., missing RNA or DNA sequencing data. This does not seem to be the case: the model only imputes the latent embedding of the missing modality via similiarity retrieval of latent embeddings. The quality of these imputations is only evaluated at the level of the representation space (Cosine Similarity), patient identities (Recall @ k), clinical annotations (HitRate) but not at the level of whether the underlying associated modality features match.
- The evaluation of the retrieval results in Fig. 3e-h misses baselines making an assessment of the performance difficult. At the moment, the study compares only against a second model instance of BLOOM-HiCLP trained with an InfoNCE instead the proposed hierarchical soft-target loss. This is rather an ablation of the method than a baseline. Possible baselines here for instance be raw features, prinicpal components of these, single modality embeddings as used in Fig. 4b-e. Moreover, it would be helpful to indicate as a lower bound the HitRate achieved by uniformly random retrieval.
- The main text of the paper specifies the actual modalities mostly only rather vaguely as "RNA/DNA-derived modalities/features". Clarifying these early on would help understandability.

Questions:
- The description of the DNA tower in the appendix states that clincial variables and disease names are incorporated into the embedding. Could the authors specify these to outrule potential information-leakage with respect to the clinical downstream tasks?

---

### Official Review · Reviewer_dWQZ · 2026-02-24
**A well-motivated paper with promising results, questions remain regarding data and evaluation pipelines**

**Rating:** 6
**Confidence:** 3

**Review:**

$\textbf{Quality}$

The paper is well-motivated with a clear presentation of the introduced methods. The proposed framework appears to be task-appropriate and should bring clear benefits. Nonetheless, several concerns about the handling of the dataset require further clarification to be fully confident in the current impact of the suggested methodology.

$\textbf{Clarity}$

The paper is overall well-presented. The methodology part appears to be clear and motivated with important details regarding the architecture and the hyper-parameters being provided in the Appendix.

However, the presentation of the dataset in main text may not be sufficiently clear for readers unfamiliar with hematological data. The current version of the Appendix A.1, which is critical to understand the link between the imposed taxonomy and the prominent factors of variation in the dataset, is similarly very condensed. This part would benefit from a more detailed presentation for a wider audience.

$\textbf{Originality and Significance}$

The proposed Hierarchical soft-target contrastive objective is rooted in prior work aimed at addressing the underutilised hard negatives in InfoNCE beyond biomedical data. This should be reflected in the paper. Examples include CLIPs: MedCLIP [(Wang et al., EMNLP'22)](https://github.com/RyanWangZf/MedCLIP), SoftCLIP [(Gao et al., AAAI'24)](https://dl.acm.org/doi/10.1609/aaai.v38i3.27955), NB-CLIP [(Wei et al., WACV 2025)](https://openaccess.thecvf.com/content/WACV2025/papers/Wei_Relaxing_Binary_Constraints_in_Contrastive_Vision-Language_Medical_Representation_Learning_WACV_2025_paper.pdf); (weakly) supervised contrastive objectives [(Dufumier et al., MICCAI 2021)](https://miccai2021.org/openaccess/paperlinks/2021/09/01/112-Paper0382.html).

Overall, however, the application to the WHO5 blood cancer classification system appears original, including the success of a relatively lightweight multimodal integration for this data. The results are promising even though further analysis of the evaluated cohort would help better understand their significance.

$\textbf{Pros}$

* **[P1]** The method is sound and well-adapted to the problem that the authors are trying to solve. The architecture naturally supports variable length inputs, enabling appropriate handling of variant- and fusion- event levels and addressing the missing modalities problem.
* **[P2]** The multimodal integration and the hierarchal objective both appear to contribute to the improved performance.
* **[P3]** The authors achieve substantial improvements at a relatively modest scale and computational cost, which is a clear advantage given the limited training data.
* **[P4]** The authors stratify the training data by institution and diagnosis.

$\textbf{Cons}$

Most of the weaknesses below are linked to the potential biases and batch effects given the current scale of ID data (IIDI samples: n=5,296). Additionally, an analysis of imbalances and coverage of labels across the hierarchical levels would make the results more convincing.

* **[C1]** The definitions and the significance of the four WHO5 levels are not clearly presented. High-level descriptions should be provided in the appendix (or even in the main text). The sample distribution per label across training and evaluation folds (at least for the top levels of the taxonomy) should be reported to illustrate data imbalances.

* **[C2]**  Several details regarding the data splits and the stratification remain unclear. a) The train/validation/test partition would benefit from a several-fold cross-validation split. It remains unclear if the difficulty of the evaluation tasks varies significantly across the subsets of the IIDI data. For instance, the retrieval metrics on Val and Test appear to be higher than on Train. b) Designation of the three OOD institutions and their similarity to the ID samples is not clear.

* **[C3]** The performance of LATTICE is not compared to InfoNCE embeddings on downstream tasks.

$\textbf{Questions}$

* **Additional comparison to InfoNCE** Can the authors provide a comparison to InfoNCE-trained representations for the downstream tasks (figure 4)?

* **The impact of the 4-level hierarchy.** Is the proposed 4-level definition of the contrastive loss the only way to interpret the WHO5 taxonomy? Have the authors tried alternative levels of granularity in the contrastive loss to contextualise the contribution of each level (e.g. vary the number of levels by merging/splitting/removing some of the existing levels). For instance, how much of the performance improvement can be explained by aligning to COOs, which appears to be a prominent axis of variation according to A.1.

* **Possible biases and imbalances.** Can the authors provide a more detailed description of how well the different levels of the imposed hierarchy are represented across the subsets of data? How do the authors assess the impact of the potential imbalances on the results of the retrieval and recall tasks?

* **Cross-validation and train/val/test splits.**  Given the amount of samples, can the authors provide results for multiple train/val/test folds within a subset of IIDI data? How were the OOD institutions defined? Do the authors have an interpretation for the stronger retrieval performance on the test subset (figure 3 (b-d))?

---

### Official Review · Reviewer_fQfN · 2026-02-25
**Adding benchmarks and experiments to enhance the study**

**Rating:** 6
**Confidence:** 4

**Review:**

The paper built a CLIP-style model that aligns RNA and DNA data into a shared embedding space using a hierarchy-aware contrastive loss.
Strengths:
- The learned embeddings transfer to real clinical tasks and remain stable under institutional shifts.
- The embeddings capture biological structure, not just paired similarity.
- Achieves good performance with far fewer trainable parameters than full fine-tuning.

Weaknesses:
- The taxonomic distance weighting is manually defined and not learned or validated. The author should add justification of the choice.
- Imputation is measured by retrieval similarity, not actual reconstruction accuracy. The authors should find a more direct way of evaluating imputation.
- No comparison to MOFA+, MOGONET, moBRCA-net, MoXGATE, VAEs, or other multi-omics integration models.
- Author can analysis of performance as paired data decreases.
- Please consider benchmarking or dicussing the relevent mult-omic approaches for diffrent cancers to position you work better. Here are few examples: https://doi.org/10.1186/s13059-020-02015-1, https://doi.org/10.1038/s41467-021-23774-w, https://doi.org/10.1186/s12859-023-05273-5, https://doi.org/10.48550/arXiv.2506.06980
- Please consider adding explanation of which genes or mutations drive similarity.

---

### Meta-Review · Area_Chair_SFna · 2026-02-28

**Recommendation:** Accept (Poster)
**Confidence:** 3

**Metareview:**

Accept

---

### Decision · Program_Chairs · 2026-03-02

**Decision:**

Accept (Poster)

**Comment:**

Please see the meta-review.